# The Pressure Compaction of Zr-Nb Powder Mixtures and Selected Properties of Sintered and KOBO-Extruded Zr-xNb Materials

**DOI:** 10.3390/ma14123172

**Published:** 2021-06-09

**Authors:** Lucyna Jaworska, Tomasz Skrzekut, Michał Stępień, Paweł Pałka, Grzegorz Boczkal, Adam Zwoliński, Piotr Noga, Marcin Podsiadło, Radosław Wnuk, Paweł Ostachowski

**Affiliations:** 1Faculty of Non-Ferrous Metals, AGH University of Science and Technology, 30-059 Krakow, Poland; skrzekut@agh.edu.pl (T.S.); mstepien@agh.edu.pl (M.S.); pawel.palka@agh.edu.pl (P.P.); gboczkal@agh.edu.pl (G.B.); zwolo@agh.edu.pl (A.Z.); pionoga@agh.edu.pl (P.N.); r.wnukk@gmail.com (R.W.); pawel.ostachowski@agh.edu.pl (P.O.); 2Łukasiewicz Research Network—Krakow Institute of Technology, 30-418 Krakow, Poland; marcin.podsiadlo@kit.lukasiewicz.gov.pl

**Keywords:** zirconium powders, SPS, severe plastic deformation, relative density, hardness, phase composition, tensile tests, corrosion resistance

## Abstract

Materials were obtained from commercial zirconium powders. 1 mass%, 2.5 mass% and 16 mass% of niobium powders were used as the reinforcing phase. The SPS method and the extrusion method classified as the SPD method were used. Relative density materials of up to 98% were obtained. The microstructure of the sintered Zr-xNb materials differs from that of the extruded materials. Due to the flammability of zirconium powders, no mechanical alloying was used; only mixing of zirconium and niobium powders in water and isopropyl alcohol. Niobium was grouped in clusters with an average niobium particle size of about 10 μm up to 20 μm. According to the Zr-Nb phase equilibrium system, the stable phase at RT was the hexagonal α-phase. The tests were carried out for materials without the additional annealing process. The effect of niobium as a β-Zr phase stabilizer is confirmed by XRD. Materials differed in their phase composition, and for both methods the β-Zr phase was present in obtained materials. A very favorable effect of niobium on the increase in corrosion resistance was observed, compared to the material obtained from the powder without the addition of niobium.

## 1. Introduction

Interest in Zr-xNb double alloys has been ongoing since the end of the 1960s [1]. Zirconium is a high-melting metal with excellent resistance to corrosive agents. It does not dissolve in acids and alkalis. However, it may self-ignite when exposed to air. Zr-1% Nb (E 110) and Zr-1% Nb-0.35% Fe-1.2% Sn (E635) alloys are the basic material used in nuclear energy in Russia as fuel elements and fuel rod arrays in active reactors [2]. Zr-2.5% Nb is the standard material for CANDU reactor (Canada Deuterium Uranium reactors) pressure tubes [3]. The Zr-1% Nb and the Zr-1% Nb-0.35% Fe-1.2% Sn alloys have high creep resistance, high mechanical properties under irradiation and are highly resistant to corrosive agents. They retain very high properties at temperatures up to 350 °C. Impurities such as C, N, F and Cl are most dangerous for the corrosion resistance of the Zr-1Nb type alloys. Zr-Nb alloys which are used in active reactors in Russia are the basis for new modifications of materials for nuclear applications [4]. At room temperature, in Zr-xNb, there is the α-phase (hexagonal close-packed, hcp) and the β phase (body-centred cubic, bcc). At room temperature, the α phase contains about 0.2% Nb because niobium has very low solubility in the α-phase, and the β-phase contains 95% Nb [5]. Al, Sn and O are stabilizers of the α-phase (hcp) of zirconium and these elements increase the allotropic transformation temperature for zirconium while β-stabilizers—e.g., Nb, Ta, Mo, Fe, and Ti—decrease the temperature of the allotropic transformation. Niobium stabilizes the (bcc) β-phase. Mo, Fe, Cr, Ta and Ti could be added for the stabilization of precipitations of beta-phase [6,7,8]. Tubes for nuclear applications, made of Zr-xNb alloys, are extruded at 820 °C, drawn to obtain high accuracy of dimensions, and then annealed to 400 °C to reduce stresses [9]. Most Zr-xNb alloys are produced by various melting and casting methods. The method of obtaining the material has a great influence on the phase composition of the material obtained. According to the phase system at room temperature, depending on the niobium content in the alloys, the alpha and beta phases of zircon should be present in an amount depending on the proportion of niobium. R. Kondo et al. confirmed that the amount of β-phase in Zr-Nb alloys increases with increasing Nb content for Zr-xNb alloys (x: 3–24 mass% Nb). Alloys in this work were arc melted from a pure Zr and Nb under an Ar atmosphere on the water-cooled copper hearth in an arc melting furnace; after that, the alloys were re-melted and cast into a sand mold using a centrifugal casting machine [10]. The tests realized by R. Kondo et al. confirmed the presence of the “high-pressure” ω-Zr phase, for the 6 mass% up to 16 mass% of niobium content. For a higher amount of Nb than 16 mass%, the as-cost Zr-Nb alloys contain β phase. For Zr-10 mass% Nb and the Zr-15 mass % Nb alloys after the solution-treating, in a quenched state or an annealed the ω-Zr phase was identified [11]. The transition pressure from α-Zr to ω-Zr is estimated as 2.10 GPa at 20 °C [12,13]. The transition of the α to ω phase (or the reverse transformation) is not well understood [14,15,16]. The ω-Zr phase can form martensitically by quenching from the β-phase to room temperature. However, the ω-phase can also form during isothermal aging heat treatment [17]. The strain inducing the ω-phase is observed during plastic deformation [18]. The content ω-phase, was maximum for the 15% or 17 mass% of Nb content in Zr-Nb alloy [19]. The phase composition of zirconium alloys significantly influences the mechanical properties of these alloys. The mechanical properties of Zr-xNb can also be controlled by thermomechanical processing [17]. The formation of ultrafine-grained microstructures in the Zr-xNb alloys improves the yield and strength of the alloy [20]. Zr-xNb are used in nuclear reactors. Additionally, the Zr-Nb binary alloys were investigated to develop a new metallic biomaterial with a low magnetic resonance susceptibility [21]. Zirconium alloys are considered a substitute metal for titanium in implant applications [22,23]. Zr-Nb alloys are preferred not only for their biological but also mechanical compatibility with the human organism. Both Zr and Nb exhibit high biocompatibility [19]. Zirconium alloys, also are good as implant candidate materials due to their bio-corrosion resistance and low magnetic susceptibility. Other materials that are used for implants, such as Co-Cr alloys, stainless steels and titanium alloys become magnetized during magnetic resonance imaging (MRI), which makes this research difficult. Implants made of zirconium alloys with niobium—thanks to lower magnetic susceptibility—disrupt the MRI image less [21]. A layer of apatite forms on the surface of the zirconium alloys implants in organisms, ensuring a good connection between the bone and the implant [24]. In 2003, a femoral head in hip arthroplasty surgery was made of OXINIUM^TM^. Since 2005 there are commercial solutions of knee implants OXINIUM™ alloy, produced by Smith & Nephew, Inc., Memphis, TN, USA [25]. This material is made from zirconium with 2.5% niobium metal and a 4 to 6-μm gradient layer of zirconium oxide on the metal surface, formed in the special oxidation process [26,27,28,29]. Zirconium alloy implants eliminate the risk to nickel-allergic and to cobalt-allergic patients [30]. The cell viability and bone-bonding characteristics of Zr-Nb system alloys are improved with increasing Nb content [31]. Currently, researches related to the use of zirconium metallic glasses for implants are carried out [24,32]. What the future holds is the possibility of producing various, even very complex shapes of products by additive methods, including those using powder sintering [33,34]. Research was undertaken in the field of the sintering zirconium powders [35,36] and powder mixtures on the base of zirconium [37]. Powders of industrial quality were used in these researches, i.e. powders with about 2.5% hafnium. Separating hafnium from zirconium is difficult because hafnium and zirconium form solid solutions with unlimited solubility in the solid phase [38]. In nuclear applications, hafnium is separated from zirconium, which significantly increases the price of materials [39]. For the nuclear application, hafnium separation should be conducted. Due to the flammability of zirconium, technological operations related to zirconium powders are difficult and require special solutions to limit zirconium contact with oxygen. The second factor that hinders technological processes is the easy oxidation of zirconium powders. Materials with a relative density in the range of 96–100% were obtained, depending on the SPS (Spark Plasma Sintering) parameters [35,37]. Materials sintered by SPS consist mostly of the α-Zr. Materials have the tendency to crack, which indicates the presence of thermal stresses generated in the cooling process. The material cooling rate should be reduced below 100 deg/min, which allows for the avoidance of cracking [37].

For obtaining study materials, presented in this paper are methods of powder metallurgy, which so far have not been used to obtain solid materials with a zirconium matrix, except for sintering zirconium metallic glasses. The methods of powder consolidation allow for the manufacturing of products with more complex shapes and reduce the finishing machining for the products. Also, an interesting issue is the possibility of obtaining scaffolds for medical applications. The first obtaining method used for materials was SPS, a non-equilibrium sintering method. Materials sintered with this method have a more complex phase composition compared to materials sintered with free sintering [40]. The second method, which was used for the zirconium powder consolidation in this work, was extrusion using KOBO method. Extrusion by KOBO method uses the phenomenon of permanent change of the deformation path during the course of the whole process, realized by cyclic, two-sided, plastic twisting of the metal [41,42]. KOBO is one of the Severe plastic deformation (SPD) methods, using very large strains and the scheme of apparatus presented in Figure 1.

The tests of selected mechanical properties were carried out for sintered materials without applied heat treatment. The corrosion resistance of zirconium obtained by melting, sintered zirconium powder and a sintered mixture of zirconium powder with niobium was also determined.

The aim of the research was to determine the effect of niobium on the phase composition, microstructure, strength properties and corrosion resistance of materials with a zirconium matrix obtained from powders. The research concerns the use of powders in the production of zirconium details, the influence of niobium on the stabilization of the β-phase and the presence of the ω-phase in sintered materials and extruded KOBO compacts.

## 2. Materials and Methods

Commercially available zirconium powders (supplier BIMOTECH, Wroclaw, Poland, grain size indicated by the producer below 60 μm, 99.9% purity) and two niobium powders (prod. Alfa Aesar, Ward Hill, MA, USA, first powder 1–5 μm, 99.8% purity and second powder with average size 44 μm, 99.9% purity) were used. The average hafnium content in zirconium is 2 mass% and was determined by the EDS method, while the oxygen content in zirconium is 1.6%, 120 ppm nitrogen, hydrogen 0.155% [35]. The niobium powders were both fine and coarse, taking into account the possible differences in their distribution in the volume of the mixtures, and, as a result, in the microstructure of the consolidated material. The assumption of the research is, inter alia, determination of the SPS and the KOBO influence on the phase composition of materials after powder consolidation. For this reason, the composition of mixtures was assumed to correspond to the previously tested melted alloys containing 1 mass%, 2.5 mass% and 16 mass% of niobium [10]. In the melted Zr-xNb alloys, the presence of the ω-Zr phase was found for the niobium content of 6–16 mass%; therefore, the tests included mixtures of zirconium with 16 mass% niobium [21]. Zirconium and niobium powders were mixed in a Fritsch Pulverisette 7 ball mill (FRITSCH GmbH, Idar-Oberstein, Germany), with zirconia grinding bowl (size 80 mL) and zirconia grinding balls (diameter of ball was 10 mm; the mass of the balls in the grinding bowl was 45 g; the powders mass was 30 g). The powders were mixed for 16 h in water and isopropyl alcohol with a mixing speed of 100 rpm. The use of higher mixing parameters is dangerous and may cause rupture of the grinding bowl. At elevated temperatures, the powders of zirconium are capable of igniting spontaneously in the air. The conditions for the preparation of mixtures with zirconium were developed in the research presented in [37]. All preparations related to the powders, their storage and mixing as well as the storage of the sintered and extruded material should be carried out with limited access to air (oxygen) due to the high reactivity of zirconium. The Zr-1 mass% Nb, Zr-2.5 mass% Nb and Zr-16 mass% Nb powder mixtures were dried under vacuum. The criterion for selecting the sintering parameters was the maximum relative density and the smallest possible grain growth of the sintered microstructure. An SPS furnace (FCT Systeme GmbH HP-D5/2, Frankenblick, Germany) was used to sinter the powder mixtures. The SPS was carried out using temperatures of 1100 °C, 1200 °C and 1300 °C, a pressure of 35 MPa and a sintering time of 1 min, in argon. The sintered and extruded materials were characterized for their density using the Archimedes’ principle. Using these criteria, an optimum temperature of 1200 °C was selected. The diameter of the sintered samples was 20 mm. The powder mixture (52 g) was placed in a Cu-ETP container (container outer diameter 40 mm, height 36 mm, wall thickness 4 mm) and then closed with a copper plug. The copper container was used in KOBO process to limit the access of air to the zirconium mixture. The extrusion process using KOBO method [41] was carried out on a laboratory hydraulic press with a maximum punch pressure of 1 MN. During the extrusion process with KOBO method, the die is twisted in reverse, which differentiates this technology from the conventional extrusion process. The batch was placed in a press container heated to a temperature of 400 °C for 30 min. The pressing was performed with a constant punch speed of 0.2 mm/s, obtaining a material with a diameter of 12 mm. During shaping, the die was twisted in reverse by an angle of ±8° with a frequency of 5 Hz. Sintered and extruded materials were ground after sintering to remove the outer layer. The samples were ground on abrasive papers using 320 to 4000 papers (gradiations of papers), and then polished with a diamond paste with a grain size of 3 microns. These operations were performed on the Roto-Pol-11 device (manufactured by Struers, Copenhagen, Denmark). A finishing treatment was carried out with the mixture of 90% OPS polishing agent with 10% hydrogen peroxide. The phase composition studies were carried out using the Bruker D8 Advanced/Discover device (Bruker, Billerica, MA, USA), equipped with a cobalt cathode lamp. For obtaining the monochromatic radiation of length λCoKa1 = 1.78897 Å, the Fe filter was used. The analysis of the phase composition of the tested materials was carried out using the PDF-2 crystallographic basis (International Centre for Diffraction Data, Newtown Square, PA, USA). The measurement was made with a scan step size of 0.02°. The following cards were used to analyze the phase composition: Zr-α: 00-005-0665, Zr-β: 00-034-0657, Zr-ω: 00-026-1399, Nb: 00-035-0789. Microstructural studies were carried out on the Hitachi SU-70 scanning microscope (Hitachi High-Technologies Corporation, Tokyo, Japan). EDS spectroscopy was used for the study of an elements distribution. Hardness measurements were carried out by the Vickers method with a load of 19.807 N, using a Shimadzu HMV-2 T microhardness tester (Shimadzu Corporation, Kyoto, Japan). The mechanical properties were determined in a tensile test carried out at temperatures of 20 °C and 400 °C. Due to the small size of samples—presented in Figure 2—they were located in special holders, which were designed for high-temperature tests to increase the accuracy of averaged values [43,44].

Machining of the samples was conducted using the BP05d electro-discharge machine (Zakład Automatyki Przemysłowej B.P., Konskie, Poland) and a WOLCUT 500 wire (Wolco Sp.z.o.o, Lublin, Poland) with a diameter of 0.25 mm and 0.15 mm. The voltage of machining was 76 V and the current was 300 mA. The electrochemical characterization was performed in the 3.5% NaCl solution. The active surface of the samples was ground on an 800 grid abrasive paper and cleaned in an ultrasonic bath for 5 min. Then, the sample was dried in a cold air stream. The convention three-electrode electrochemical cell with the Ag/AgCl (3M KCl) as a reference electrode and platinum plate as the counter electrode was applied. The specimens were mounted in a PTFE holder. The potentiodynamic polarization scan was performed from −0.25 V vs OCP to potential 1 V vs. reference electrode or the current density reaches 20 mA/cm^2^ (depending on which condition is met first). Prior to the polarization test, each sample was conditioned for 1 h in the solution to obtain an open circuit potential. All experiments were carried out at room temperature (22 ± 2 °C, air-conditioned room). The corrosion current was estimated from the Tafel slopes. The experiments were repeated at least three times to check the reproducibility of the result. For a comparative analysis of the corrosion resistance, zirconium foil, 0.711 mm thick (Alfa-Aesar, annealed, 99.5% metals basis excluding Hf), was used.

## 3. Results

The process of mixing zirconium and niobium powders was used. Mechanical alloying was not undertaken for fear of the explosive tendency of zircon. Zirconium is stored in water. Water protects the zirconium powder against oxidation and the possibility of spontaneous combustion; therefore, water was selected as the medium in which the powder mixing process was carried out. Additionally, isopropyl alcohol was used. The powders were dried in a vacuum prior to the sintering and extrusion process. In the mixing process, ZrO_2_ grinding bowl with ZrO_2_ balls were used. ZrO_2_ was selected because of its relationship to zirconium, and thus the proportion of other impurities in the obtained material was limited. However, due to the relatively low density of zirconium oxide of 5.68 g/cm^3^, mechanical alloying is difficult due to the low process energy compared to containers and grinders made of steel or tungsten carbide.

### 3.1. The Powders Consolidation

Results of the SPS process for Zr-xNb mixtures prepared in isopropyl alcohol are presented in Table 1 and Table 2.

Results of the SPS process for Zr-xNb mixtures prepared in water are presented in Table 3 and Table 4.

The influence of the medium intended for mixing the powders slightly affects the density of the sintered materials. In Table 5 the parameters of KOBO extrusion process and relative densities and hardness measurements are presented.

For the E110 (Zr-1 mass% Nb) alloyed zirconium consists of the α-Zr phase, microhardness (for 100 g loading) is 149 ± 5 [45]. In sintered materials, the high hardness values were influenced by thermal stresses resulting from the high cooling rate in the SPS process—50 °C per min. The tests were carried out for materials that were not subjected to heat treatment (to reduce stress). The hardness of sintered Zr-xNb increased with the increase in niobium participation. The average hardness values for sintered materials with the same niobium content were similar regardless of the size of the niobium particles in powders. For 16 mass% of niobium participation the standard deviation was higher than for 1% and 2.5% of niobium because of differences in hardness of these metals. Hardness for the KOBO-extruded samples was higher than for sintered materials. In this case, the deformation of the samples in the extrusion process generated stresses, and as a result, strainhardened materials were obtained.

### 3.2. Microstructures of Sintered and Extruded Zr-xNb Materials

In the microstructure of the SPS Zr-2.5 mass% Nb presented in Figure 3A,B, there were lentoid and isometric grains, and the microstructure was similar to the SPS-sintered Zr without additives (containing the α-phase) [35]. Large clusters of needles were concentrated around the niobium inclusions. This area with needles was characterized by the presence of a higher concentration of niobium. In the case of SPS Zr-16 mass% Nb, shown in Figure 3E, the needle structure around the niobium disappeared, and the lenticular microstructure predominated. Niobium was evenly distributed over the entire volume of all sintered materials. In Figure 3B, the shape of niobium inclusions is slightly elongated, which was a result of applied pressure in the SPS process; the inclusions were approximately 10–20 μm in diameter.

The extruded material microstructure presented in Figure 3C,D and in Figure 3F is different, as there was no lenticular structure and there were larger clusters of niobium grains. For comparison, the microstructure of cast materials is plate-like, which has been characterized by S. Cai et al. [46]. For both the SPS and extruded materials, the pores were mainly found in the areas surrounding the niobium particles.

In the Zr-5 mass% Nb alloys (melted materials), the α-phase had a plate-like shape with the β-phase distributed between the α-grains [46] or at the grain boundaries for Zr-2.5 mass% Nb [47]. With the increase in proportion of niobium in the material, more of the β phase appeared, and the grains stayed more isometric because of the cubic structure of this phase. In cast materials, the distribution of niobium is uniform and depends on its solubility in the zirconium phases. Heat treatment changes the phase composition of melted zirconium alloys [21].

### 3.3. Phase Coposition of Sintered and Extruded Zr-xNb Materials

The starting powders consisted of the α-Zr. The SPS sintered zirconium material without niobium addition consisted of the α-phase. In sintered materials with 2.5 mass% Nb—presented in Figure 4—the dominant phase was the α phase. Niobium peaks are visible for sintered materials and the KOBO-extruded materials. For extruded materials with the addition of 2.5 mass% of niobium, a small amount of the β phase appeared, which is presented in Figure 4. The intensity of the β phase increased with the increase of the Nb content. More of the β phase was in the sintered SPS Zr-16mass%Nb, which is visible in Figure 5. In the tested materials, niobium was not evenly distributed throughout the entire volume of the material, as is the case in melted materials [21]. The materials had areas with a high concentration of niobium in micro-areas, and in large areas without the presence of niobium. This affects the phase composition of sintered and extruded materials and results in a limited content of the β phase compared to melted Zr-xNb alloys.

Figure 4 shows the XRD diffractograms of Zr-2.5 mass% Nb sintered and extruded materials. The intensity of peaks originating from the α phase for the SPS materials decreased with the increase of the Nb content, which is visible in Figure 4 and Figure 5. The β phase and ω phases were observed in the SPS Zr-2.5 mass% Nb and the SPS Zr-16 mass% Nb materials. For the extruded Zr-16 mass% Nb, the α phase was still the dominant phase. The β phase participation in the extruded materials was definitely lower than in the sintered materials, obtained at a high temperature of 1200 °C. At this temperature, the niobium diffusion phenomenon occurred, as shown in Figure 3b, which affected the stabilization of the β phase.

There was an unknown single peak in the diffractograms between 33–35° (Figure 3 and Figure 4) that has not been determined; this peak can be found in the diffraction patterns in other publications and the peak was not described [10,19,45].

### 3.4. Mechanical Properties Tests of Sintered and Extruded Zr-xNb Materials

The tensile tests were carried out for samples after extrusion, presented in Figure 6 and Figure 7 and after sintering, presented in Figure 8. The materials are characterized by brittle cracking at room temperature but also at the temperature of 400 °C, with the exception of pure zirconium sintered using the SPS method, at 400 °C, which was the reference sample in these tests. The cause of the brittle fracture of the samples is the thermal stresses for the samples produced by the SPS method, and for the extruded samples—deformation stresses. Properties for melted zirconium alloys [1,8] were carried out after heat treatment or recrystallization. At room temperature, the tensile strength of the annealed pure melted zirconium was below 300 MPa and increased with increasing niobium content [8].

In the tests presented in Figure 6, Figure 7 and Figure 8, the stresses were related to the starting cross-section of the samples.

Table 6 presents the values of ultimate tensile strengths for investigated materials.

Materials obtained using zirconium powders with niobium participation had a lower tensile strength value than pure sintered or extruded zirconium powders without additives, at room temperature (135 MPa for the sintered Zr, 88 MPa for the extruded Zr) and at 400 °C (285 MPa for the sintered Zr, 377 MPa for the extruded Zr). Materials obtained from powders with niobium have a lower tensile strength value than pure zirconium without additives and the tendency of increasing the strength value and strains with the increase in the proportion of niobium. A very important factor influencing the mechanical properties of materials obtained from powders is porosity and non-uniform distribution of niobium, which is also present in the materials in unreacted form. The mechanical properties for melted zirconium alloys [1,8] were carried out after heat treatment or recrystallization. Tensile strength of the annealed pure melted zirconium, at room temperature, was below 300 MPa and increases with increasing niobium content [8]. In Table 6 the values of ultimate tensile strengths are presented. Materials obtained using zirconium powders with niobium participation have a lower tensile strength value than pure sintered or extruded zirconium powders without additives, at room temperature (135 MPa for the sintered Zr, 88 MPa for the extruded Zr) and at 400 °C (285 MPa for the sintered Zr, 377 MPa for the extruded Zr). Materials obtained from powders with niobium have a lower tensile strength value than pure zirconium without additives, but the tendency of increasing the strength value and strains with the increase in the proportion of niobium. A very important factor influencing the mechanical properties of materials obtained from powders is porosity and non-uniform distribution of niobium, which also is present in the materials in unreacted form.

Figure 9a,b and Figure 10a,b show the fracture surfaces of the Zr-16 mass% Nb samples, for tensile tests carried out at 20 °C and 400 °C, respectively. For tests carried out at 20 °C, regions of brittle transcrystalline and brittle intercrystalline fracture are visible in Figure 9a,b. The material is characterized by a brittle fracture. At a temperature of 400 °C, the fracture surfaces shown in Figure 10a,b are characterized by a presence of larger ridges of deformed material which are due to improved ductility of the zirconium phase at a higher temperature. This tendency is shown by all tested materials. This behaviour of zirconium and its alloys at room and high temperatures is typical and confirmed in studies by other researchers [48].

The tensile test curve at 400 °C for the SPS sintered zirconium without additives is presented in Figure 8. This curve is representative of ductile materials, while other sintered and extruded materials with niobium and KOBO-extruded zirconium without niobium still behave like brittle materials.

In most cases of melted alloys, the process of plastic working and subsequent heat treatment is carried out. In melted, rolled and heat-treated Zr-xNb alloys, the solid-solution, grain size and niobium precipitation are responsible for the high strength value [8]. In these studies, the researchers focused on the effect of consolidation methods on phase composition and properties immediately after consolidation processes. In general, the tensile strength values for both SPS and KOBO extrusion materials do not differ significantly, as shown in Table 6.

### 3.5. Potentiodynamic Polarization Curves for KOBO-Extruded Materials

Selected potentiodynamic polarization curves for KOBO-extruded materials, realized in 3.5% NaCl solution, are shown in Figure 11.

Materials obtained from powders consolidated with the KOBO method were selected for the corrosion study due to their higher value of relative density in comparison to the materials obtained using the SPS method, of which values are given in Table 7.

The estimated current and potential describing the corrosion behaviour of the samples are presented in Table 7 (rest potential—E_rest_, corrosion potential E_corr_, breakdown potential E_br_, corrosion current density i_corr_ and passive current density i_p_).

KOBO-extruded Zr powders without niobium additive have the worst corrosion resistance of all tested materials. The corrosion current density for this sample is 4100 nA/cm^2^. In Figure 11, the typical plateau is not visible in the passive state for potentiodynamic polarization curves of the extruded materials, unlike the reference Zr sample. Because there is no plateau, it is not possible to calculate the passive current density ip. Similar behavior was observed for the Ti materials obtained via the SPS method from powders [49]. Zirconium and titanium belong to the same group of the periodic table and have similar properties. The available literature does not provide information on corrosion tests for zirconium materials obtained from powders. The addition of Nb to the zirconium greatly increases the corrosion resistance, and the samples show more noble E_corr_ and E_rest_ compared to the reference sample.

The i_corr_ obtained for extruded Zr-16mass%Nb reached a level of approximately 70 nA/cm^2^ and did not differ significantly from the Zr reference sample, but the i_corr_ for the extruded Zr-2.5 mass% Nb was twice as large. Most likely, it resulted from the presence of Nb_2_O_5_. This oxide is an inhibitor in the exchange of electrons between the alloy and the environment [50]. The remaining parameters, gathered in Table 7, assessing the corrosion resistance of KOBO Zr-16 mass% Nb and KOBO Zr-2.5 mass% Nb, were significantly different. However, it is worth paying attention to the oscillation appearing just before the breakdown potential for KOBO Zr-16 mas% Nb sample, which may suggest its increased pitting tendency to the sample with a lower addition of Nb. Comparing E_br_ for all examined samples, we can assume that all KOBO-extruded materials have similar breakdown potential, highly lower than the reference Zr. This limits the potential application of these kinds of alloy in environments with corrosive potentials below 250 mV vs. Ag/AgCl. This may be due to the porous structure of the material. Treating titanium alloys synthesized by the SPS method [49] as reference material, significantly lower currents were obtained for Zr and Zr-xNb materials; however, they are characterized by a lower breakdown potential.

## 4. Discussion

For most commercial zirconium alloys obtained using melting methods, the basic phase is the α-phase (hcp). Niobium is a β-phase (bcc) stabilizer in zirconium alloys [5,8]. Zr-xNb alloys with a niobium content of more than 6% (up to 22 mass%) contain the ω phase (hpc). Omega zirconium phase is brittle and weakens mechanical properties of zirconium alloys, and this phase has a higher magnetic susceptibility than α and β phases [10]. The β-phase is the dominant phase in melted Zr-xNb alloys for niobium content from 9–24 mass% [10]. Zirconium materials containing β phase are characterized by good mechanical properties. For melted zirconium alloys, the annealing is conducted near the monotectoid temperature of 620 ± 20 °C [5,11,47]. The use of annealing allows for the reduction of thermal or mechanical stresses arising in technological processes. Heat treatments change the phase composition of the alloys and thus their properties. For this reason, research is also carried out for as-cast materials without annealing [10]. An increase in the share of the ω-Zr phase was found after a heat treatment for melted Zr-xNb alloys [21].

The use of powder metallurgy methods allows for production elements with sophisticated shapes, and with limited finishing machining. A metal with ideal properties similar to human bone is still in demand. One of the basic materials used for implants is titanium. For titanium, research has been undertaken on the sintering (SPS) of titanium powders [51]. Sintered zirconium can replace titanium more successfully [22]. However, with the high melting point of zirconium, its resistance to corrosive agents and therefore other types of applications related to these properties must be taken into account. Working with zirconium powders is difficult due to its flammability and reactivity with oxygen. In presented works, by applying both SPS and KOBO consolidation methods, materials with high relative density values were obtained for extruded materials up to about 98%. The tests were carried out for materials that were not subjected to heat treatment. Niobium is evenly distributed over the entire volume of all sintered materials. Irrespective of the particle size of the niobium powder and the type of liquid in which the zirconium and niobium powders were mixed for 16 h, the niobium particle size (1–5 μm, average 44 μm) after consolidation was below 20 μm. Niobium was evenly distributed throughout the entire volume of the material and was located on the grain boundaries, which had an impact on SPS process because the proportion of niobium in the micro-areas is greater than the phase fraction of niobium in the material, which influences the phase composition of consolidated materials. In these studies, the β phase and ω phases were observed in the SPS Zr-2.5 mass% Nb and the SPS Zr-16 mass% Nb materials. For extruded Zr-2.5 mass% Nb and Zr-16 mass% Nb, the α phase was still the dominant phase. The melted materials for 2.5% niobium consisted of the α-Zr phase [10]. Powder-consolidated Zr-xNb materials should be considered as composite materials. Hardness of Zr-xNb obtained by SPS and KOBO methods are about two times higher than for the melted Zr-1mass%Nb, but is very similar as for the Zr-1 mass% alloy obtained by the high-pressure torsion process, which consists of ω-Zr phase [45]. It is the result of strengthening due to thermal stresses during cooling and mechanical stresses resulting from deformation. Obtained materials are brittle at room temperature. At 400 °C, the fractures show a partially ductile fracture. The presence of residual porosity weakens the mechanical properties of the materials, which may not necessarily be a disadvantage in biological applications. The tests were carried out for small samples, in tests. It is difficult to compare the ultimate tensile strength values with other tensile tests in other publications, as there is no data for materials obtained from powders with residual porosity. In the paper [10], the pure zirconium ultimate tensile test strength is 388 MPa, for the Zr-3 mass% Nb is 786 MPa and for the Zr-16 mass% Nb is 686 MPa. The effect of the phase composition of materials and niobium participation on the strength values is clearly visible in these studies [10]. In other work, ultimate tensile strength for Zr-2 mass% Nb is about 400 MPa [8]. Ultimate tensile strengths decrease with increasing Nb content. In presented tests, ultimate tensile strengths at RT are in the range 75–87 MPa.

The corrosion tests that have been carried out indicate that some of the results obtained are better than for SPS sintered titanium powders [49]. The results of the corrosion resistance tests of materials obtained from zirconium-niobium powders indicate a high potential for their application in corrosive environments.

In future research, the obtaining of Zr-xNb alloy powders, currently not available from distributors, should be considered. The use of alloy powders will ensure an even distribution of niobium and better control of the phase composition of materials. Strength properties of materials obtained from powders should be improved by annealing. However, the application of Zr-xNb materials should be taken into consideration.

## 5. Conclusions


The tests showed no significant influence of the presence of water or isopropyl alcohol on the homogeneity of Zr-xNb mixtures. Mixing was used in the tests, and mechanical alloying was not carried out due to the tendency of zircon to ignite.The Zr-xNb powder mixture’s consolidation technique has an impact on the material’s microstructure and phase compositions. The β phase and ω phases were observed in the SPS Zr-2.5 mass% Nb and the SPS Zr-16 mass% Nb materials. For the extruded Zr-16 mass% Nb, the α phase is still the dominant phase. The beta phase participation in extruded materials is definitely lower than in sintered materials, obtained at a high temperature of 1200 °C. At this temperature, the niobium diffusion phenomenon occurs, which affects the stabilization of the β phase. The materials have areas with a high concentration of niobium in micro-areas, and areas without the presence of niobium. This affects the phase composition of sintered and extruded materials and results in a limited content of the β phase compared to melted Zr-xNb alloys.Porosity and hardening affect the tensile strength values of sintered and extruded materials. These values are lower than for alloys obtained by melting methods. However, the values known from the literature are determined for alloys after heat treatment, which reduces the stress or affects the recrystallization process of zirconium alloys. In this work, research was carried out on materials after sintering and extrusion without heat treatmentThe corrosion resistance tests were carried out for samples with the highest values of relative density. The corrosion resistance tests showed a positive effect of niobium on the values of estimated currents and potentials describing the corrosion behavior of the samples. Values for niobium materials were lower than for pure melted and heat-treated reference zirconium. However, research shows the potential of using powder techniques, e.g sintering (SPS, SLS) and extrusion, provided that the niobium will be better distributed, for example, by preparing finished Zr-xNb alloy powders.


## Figures and Tables

**Figure 1 materials-14-03172-f001:**
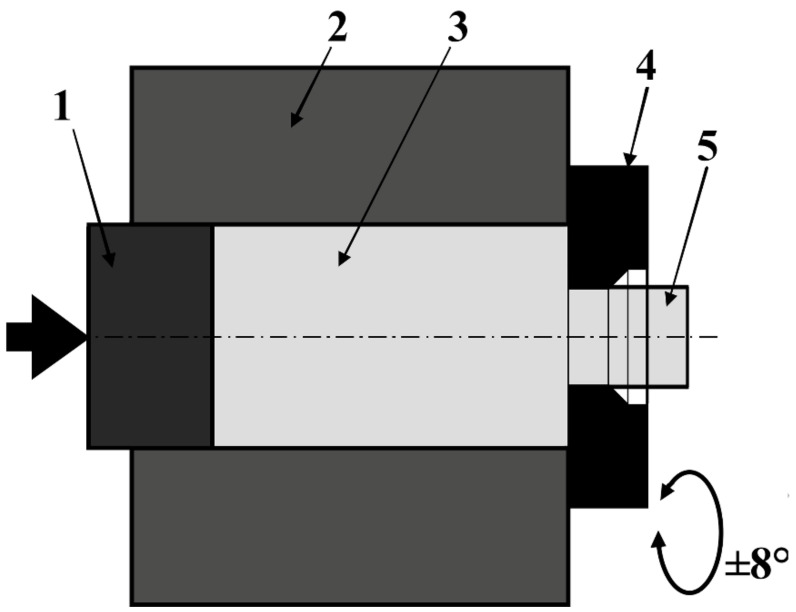
Scheme of KOBO extrusion method: 1—punch, 2—container, 3—billet, 4—reversibly rotating die, 5—extruded material.

**Figure 2 materials-14-03172-f002:**
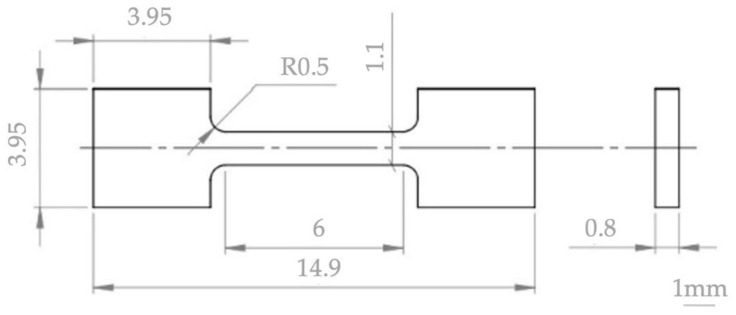
Geometry of sample for tensile tests.

**Figure 3 materials-14-03172-f003:**
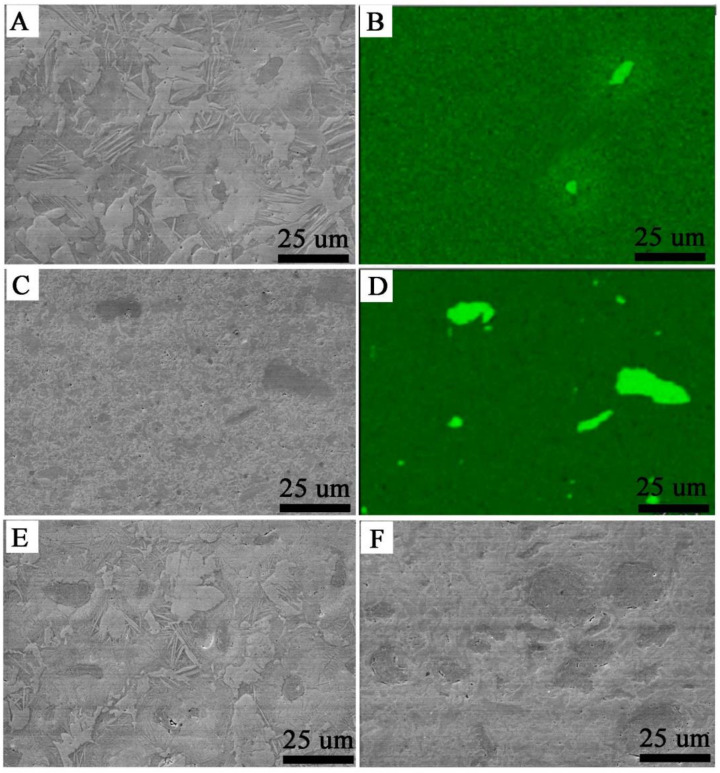
Microstructures of the SPS Zr-2.5 mass% Nb, SEM (**A**); SPS Zr-2.5 mass% Nb, the niobium distribution, EDS (**B**); KOBO-extruded Zr-2.5 mass% Nb, SEM (**C**); KOBO-extruded Zr-2.5 mass% Nb, the niobium distribution, EDS (**D**); the SPS Zr-16 mass% Nb, SEM (**E**); KOBO-extruded Zr-16 mass% Nb, SEM (**F**).

**Figure 4 materials-14-03172-f004:**
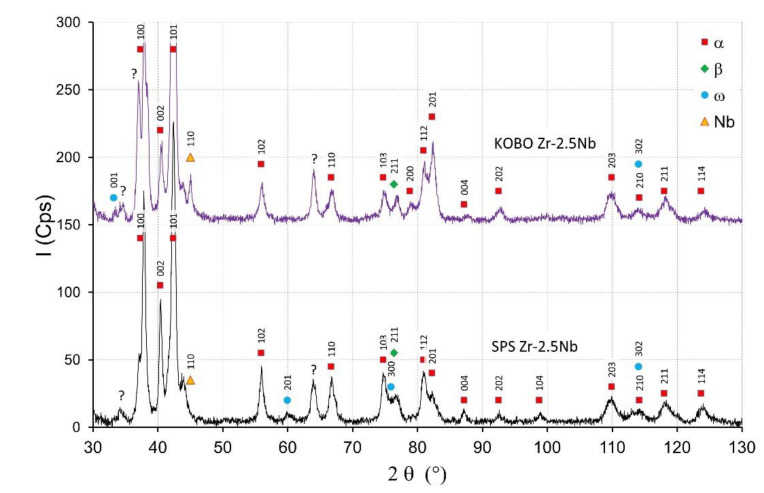
The X-ray diffractograms of the Zr-2.5mass%Nb materials obtained by SPS and KOBO-extrusion.

**Figure 5 materials-14-03172-f005:**
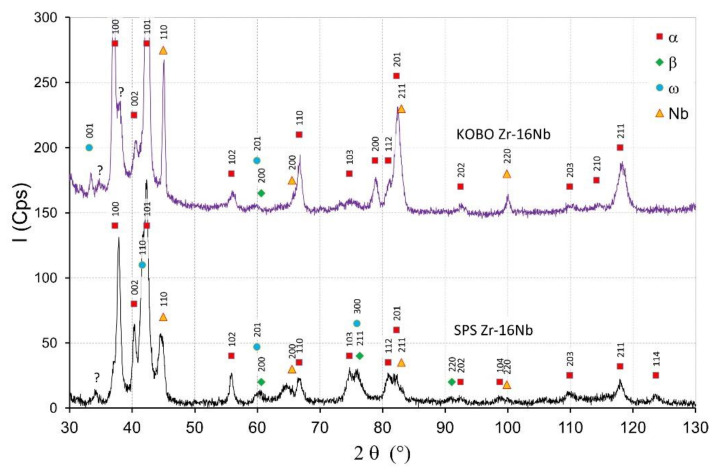
X-ray diffractograms of the Zr-16 mass% Nb materials obtained by SPS and KOBO extrusion.

**Figure 6 materials-14-03172-f006:**
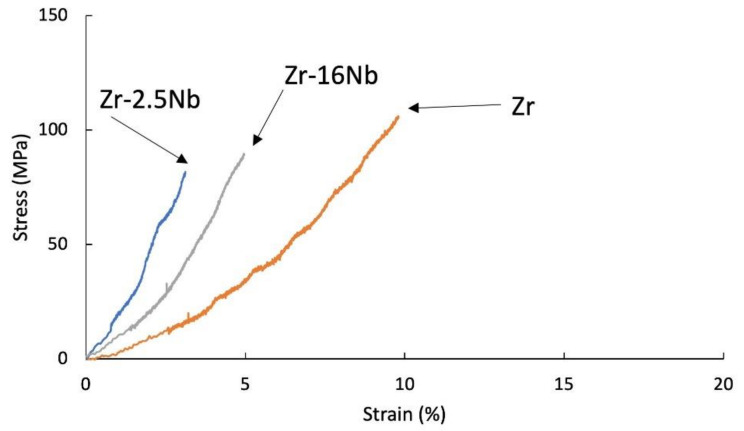
Tensile stress-strain curves of KOBO-extruded Zr-2.5 mass% Nb, Zr-16 mass% Nb and Zr materials; tests were carried at room temperature.

**Figure 7 materials-14-03172-f007:**
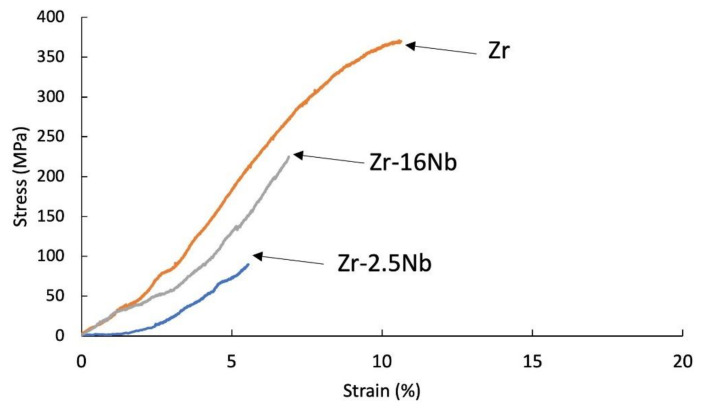
Tensile stress-strain curves of KOBO-extruded Zr-2.5 mass% Nb, Zr-16 mass% Nb, Zr materials; tests were carried at 400 °C.

**Figure 8 materials-14-03172-f008:**
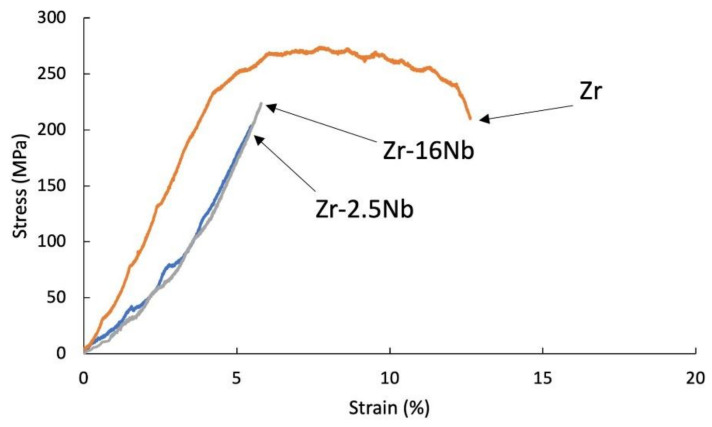
Tensile stress-strain curves of the sintered Zr-2.5 mass% Nb, Zr-16 mass% Nb and Zr materials; tests were carried at 400 °C.

**Figure 9 materials-14-03172-f009:**
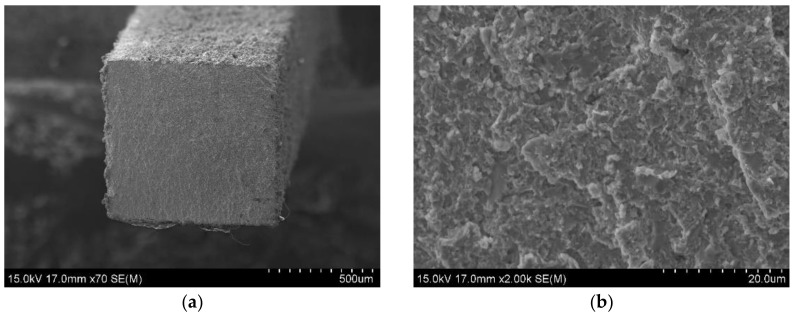
SEM observation on fracture surfaces of KOBO extruded Zr-16 mass% Nb material, tensile test was carried at 20 °C; (**a**) the fracture surface; (**b**) the magnification of 2000x, SEM.

**Figure 10 materials-14-03172-f010:**
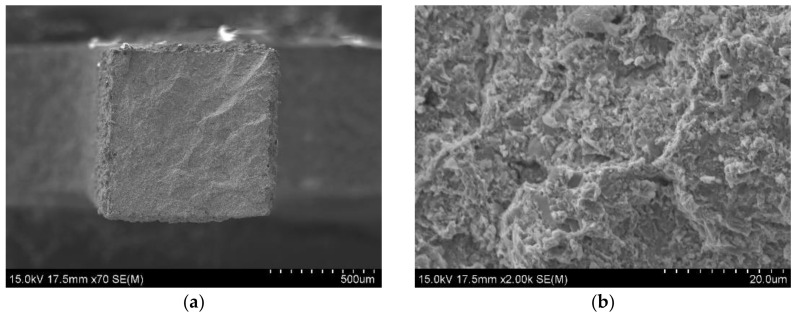
SEM observation on fracture surfaces of the KOBO extruded Zr-16 mass% Nb material, tensile test was carried at 400 °C; (**a**) the facture surface; (**b**) the magnification of 2000x, SEM.

**Figure 11 materials-14-03172-f011:**
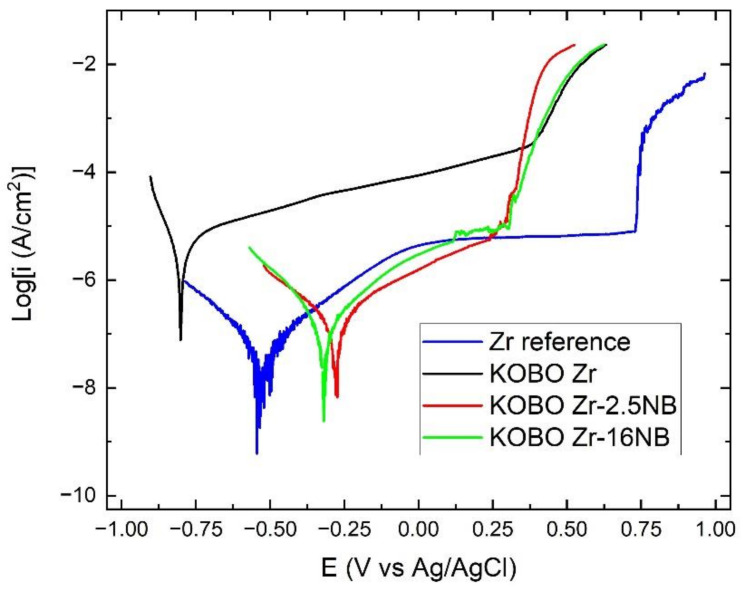
Potentiodynamic polarization curves for the corrosion test of KOBO-extruded Zr-xNb alloys in 3.5% NaCl solution.

**Table 1 materials-14-03172-t001:** The SPS process parameters, relatively densities, hardness of the sintered Zr-xNb materials obtained from mixtures with the 1–5 μm niobium powder, milled in isopropyl alcohol.

Zr-xNb	Sintering Temperature (°C)	Duration of the Sintering (min)	Relative Density (%)	Hardness * (HV2)	Standard Deviation
Zr1mass%Nb	1200	1	94.82	339	±3.77
Zr2.5mass%Nb	1200	1	94.96	377	±4.03
Zr16mass%Nb	1200	1	95.11	393	±27.21

* average value for ten indentations.

**Table 2 materials-14-03172-t002:** The SPS process parameters, relatively densities, hardness of the sintered Zr-xNb materials obtained from mixtures with the 44 μm average size of niobium powder, milled in isopropyl alcohol.

Zr-xNb	Sintering Temperature (°C)	Duration of the Sintering (min)	Relative Density (%)	Hardness * (HV2)	Standard Deviation
Zr1mass%Nb	1200	1	96.45	357	±2.23
Zr2.5mass%Nb	1200	1	96.27	371	±1.89
Zr16mass%Nb	1200	1	95.49	397	±27.12

* average value for ten indentations.

**Table 3 materials-14-03172-t003:** The SPS process parameters, relatively densities, hardness of the sintered Zr-xNb materials obtained from mixtures with the 1–5 μm niobium powder, milled in water.

Zr-xNb	Sintering Temperature (°C)	Duration of the Sintering (min)	Relative Density (%)	Hardness * (HV2)	Standard Deviation
Zr1mass%Nb	1200	1	95.75	351	±1.43
Zr2.5mass%Nb	1200	1	95.08	388	±3.94
Zr16mass%Nb	1200	1	95.31	394	±19.18

* average value for ten indentations.

**Table 4 materials-14-03172-t004:** The SPS process parameters, relatively densities, hardness of the sintered Zr-xNb materials obtained from mixtures with the 44 μm average size of niobium powder, milled in water.

Zr-xNb	Sintering Temperature (°C)	Duration of the Sintering (min)	Relative Density (%)	Hardness * (HV2)	Standard Deviation
Zr1mass%Nb	1200	1	94.97	357	±2.13
Zr2.5mass%Nb	1200	1	95.49	375	±3.31
Zr16mass%Nb	1200	1	95.07	382	±15.93

* average value for ten indentations.

**Table 5 materials-14-03172-t005:** The KOBO extrusion process parameters, relatively densities, hardness of the extruded Zr-xNb materials obtained from mixtures with the 1–5 μm niobium powder, milled in isopropyl alcohol.

Zr-xNb	Temperature of the Extrusion (°C)	Duration of the Extrusion (min)	Relative Density (%)	Hardness * (HV2)	Standard Deviation
Zr	400	0.2	97.95	397	±12.12
Zr2.5mass%Nb	400	0.2	97.31	425	±7.84
Zr16mass%Nb	400	0.2	97.84	410	±8.92

* average value for ten indentations.

**Table 6 materials-14-03172-t006:** Ultimate tensile strengths for sintered and extruded Zr-xNb materials.

Material/ * Manufacturing Process	Test Temperature (°C)	Ultimate Tensile Strength ** (MPa)	Standard Deviation
Zr/SPS	RT	135	±14.07
Zr/SPS	400	285	±21.59
Zr/KOBO	RT	88	±15.15
Zr/KOBO	400	377	±21.25
Zr-2.5Nb/SPS	RT	87	±25.37
Zr-2.5Nb/SPS	400	215	±31.03
Zr-16Nb/SPS	RT	85	±7.09
Zr-16Nb/SPS	400	218	±32.35
Zr-2.5Nb/KOBO	RT	75	±13.13
Zr-2.5Nb/KOBO	400	84	±8.04
Zr-16Nb/KOBO	RT	78	±6.41
Zr-16Nb/KOBO	400	210	±16.69

* The value of ultimate tensile strength is the mean of the measurements for the five samples. ** For extrusion tests, materials were cut in the direction of extrusion.

**Table 7 materials-14-03172-t007:** Estimated currents and potentials describing the corrosion behaviour of the samples.

Sample	E_rest_ (mV)	E_corr_ (mV)	i_corr_ (nA/cm^2^)	i_p_ (nA/cm^2^)	E_br_ (mV)
Zr reference *	−511 ± 78	−572 ± 46	61 ± 25	4576 ± 427	745 ± 50
KOBO Zr	−731 ± 40	−792 ± 42	4104 ± 248	–	338 ± 17
KOBO Zr-2.5Nb	−301 ± 29	−319 ± 19	156 ± 15	–	279 ± 27
KOBO Zr-16Nb	−347 ± 24	−357 ± 40	70 ± 32	–	293 ± 12

* Zr reference this is zirconium foil, prod. Alfa-Aesar, annealed, 99.5% metals basis excluding

## Data Availability

The data presented in this study are available on request from the corresponding author.

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
