# Peer review of "The Pressure Compaction of Zr-Nb Powder Mixtures and Selected Properties of Sintered and KOBO-Extruded Zr-xNb Materials"

_materials, 2021, doi:10.3390/ma14123172_

Round 1

Reviewer 1 Report

Dear Authors,

I have read your paper "The pressure compaction of Zr-Nb powder mixtures and selected properties of sintered and KOBO-extruded Zr-xNb materials" carefully.

This paper describes the operation properties of the Zr-xNb materials which were fabricated by mean of high concentrated energy treatments..   

The paper is easy to read.

But the methods are not properly described, so that other research groups may not reproduce them.

The paper is interesting. However, it requires few corrections.

  1. The introduction has no information about the purpose of this work. It’s recommended to add the paragraph with the purpose of this work to introduction.
  2. Please, add information about the weight of the balls and of the milling powders. 
  3. Please, add more information about the ? phase on the figure 3.
  4. Please specifically discuss the advantages of your work. Mark the main advantages compared to the other scientists (other methods). Now the discussion is poor.

The paper can be accepted for publication only after major improvements.

Reviewer 2 Report

The submitted manuscript entitled ‘The pressure compaction of Zr-Nb powder mixtures and selected properties of sintered and KOBO-extruded Zr-xNb materials’ deals with the production and investigations (microstructural, mechanical, corrosive) of Zr based materials alloyed by Nb. The manuscript is interesting and worth publishing; however, during its review a list of issues – as listed below – arose.

- Please provide an official e-mail address (instead of commercial) for all the Authors, but at least for the corresponding Author.

- What is the reason behind the 1 m%, 2.5 m% and 16 m% Nb content (the latter being one magnitude higher than the previous ones)? The higher Nb content is out of the current applications, mentioned in the Introduction.

- Table 1: what is the reason behind the different average diameter Nb particles?

- Please always let a space between the value and its unit, except in the case of ‘°C’ and ‘%’.

- A sketch about the extrusion process and its steps would be useful.

- Fig 1 is technically incorrect: (i) the longitudinaly symmetry line is missing (ii) the sample thickness is missing, (iii) the dimensioning is incomplete. Please indicate the direction of the extrusion.

- Please use subscripts in ‘ZrO2’ expression.

- Last rows of tables 2 – 5: please add the standard deviation even if it is high.

- The XRD diffractograms (figs 3 and 4) contain unidentified peaks. What are these peaks? What is the reason behind?

- ‘The cause of the brittle fracture of the samples is the thermal stresses for the samples produced by the SPS method, and for the extruded samples - deformation stresses.’ – is it possible to decrease the additional (residual) stresses by proper heat-treatment?

- Figs 5 – 7: please indicate whether true or engineering system was used. Are the curves average curves (if so, please add scatter bands) or individual curves (please indicate the number of tested samples). Please provide the photographs of the broken samples as subfigs.

- Footnote to table 7 mentions 5 samples for each condition. Please add standard deviations.

- English is not the native language of this Reviewer, but proofreading is strongly recommended.

Reviewer 3 Report

  1. On introduction, the last sentences of the research concerns, make a separate sentence with space from the beginning. Also argue the novelty of the paper.
  2. On the introduction discuss more about specific applications of this kind alloys. If are biocompatible, where can be used?
  3. Add more new references related to the field (2021, 2020)
  4. Check Table 1 and put in the square brackets the units’ measurements.
  5. Unbold the Figures and Tables mention in the text to be uniform with all text.
  6. Check Table 4, is other format.
  7. Check Line 193, 194 and again in all the text, the chemical formula for zirconia: ZrO2
  8. On the table 8, check the units measurements: ip [nA/cm2] , icorr [nA/cm2].
  9. On the Figure 3 and 4 add Miller indices.

Round 2

Reviewer 1 Report

Dear Authors,

I have read your modified paper "The pressure compaction of Zr-Nb powder mixtures and selected properties of sintered and KOBO-extruded Zr-xNb materials" carefully.

The materials and methods are properly described, so that other research groups may reproduce them. Explanations are clear and the paper is easy to read.

I can recommend the Editor to accept this revised manuscript to be published in Materials

Author Response

Thank you very much for your constructive comments and acceptance of our article, kind regards Lucyna Jaworska

Reviewer 2 Report

Thank you for all the corrections and explanations.

This Reviewer has only one remaining concern: fig. 2 is still incorrect technically: (1) the whole length of the sample is missing, (2) dashed lines in the side view are missing.

Author Response

Dear Reviewer, we have changed figure 2, thank you for your comments, kind regards Lucyna Jaworska

Reviewer 3 Report

Article is very improved. Can be accepted to be published. 

Author Response

Dear Reviewer, thank you for accepting our article, kind regards Lucyna Jaworska